# Association between purchasing behaviors and cigar use: A longitudinal analysis of Waves 1-3 of the Population Assessment of Tobacco and Health (PATH) Study

**Jessica L. King** [1]*, **Lingpeng Shan**[2], **Sunday Azagba**[2]

**1** Department of Health, Kinesiology, & Recreation, University of Utah, Salt Lake City, Utah, United States of America, **2** Department of Family and Preventive Medicine, University of Utah School of Medicine, Salt Lake City, Utah, United States of America

* jess.king@utah.edu

## Abstract

### Introduction

Over 120 US jurisdictions have implemented policies mandating minimum cigar pack quantities, yet little empirical research exists on the relationship between pack quantity and use. We examined whether cigar use was associated with purchasing cigars by the box/pack or as singles, purchase quantity, and price paid per cigar.

### Methods

Data are from Waves 1–3 (2013–2016) of the Population Assessment of Tobacco and Health (PATH) Study, analyzed in 2019. The sample included adults who reported current use of any type of cigars (cigarillos [N = 3,051], traditional cigars [N = 2,586], and filtered cigars [N = 1,295], including with marijuana) at Wave 1. For each cigar type, a generalized estimating equation model was used to examine the population-averaged effects of purchasing behavior on cigar use.

### Results

Cigar users of each type who purchased by the box or pack smoked more per day than users who purchased singles (cigarillos: β = 1.02, p<0.0001; traditional cigars: β = 1.40, p<0.0001; filtered cigars: β = 2.55, p<0.01). Cigar users who purchased larger quantities smoked more per day (cigarillos: β = 0.16, p<0.0001; traditional cigars: β = 0.04, p<0.0001; filtered cigars: β = 0.24, p<0.0001). Higher price per cigar was significantly associated with smoking fewer traditional cigars (β = -0.12, p<0.01) and filtered cigars (β = -0.86, p = 0.02), but not cigarillos (β = 0.08, p = 0.62).

### Conclusions

Smaller pack quantities and higher price per cigar were associated with smoking fewer cigars per day. Given the authority of the Food and Drug Administration and local

**Data Availability Statement:** Data are available from https://www.icpsr.umich.edu/icpsrweb/NAHDAP/studies/36498/datadocumentation#.

**Funding:** The author(s) received no specific funding for this work.

**Competing interests:** The authors have declared that no competing interests exist.

jurisdictions over cigar pack quantity, this study provides data pertinent to potential minimum and maximum package quantity regulations and policies.

## Introduction

While cigarette use declined between 2000 and 2015, cigar use overall remained stable, and increased among some populations. [1] In 2018, 3.8% (9.3 million) of U.S. adults reported current cigar use. [2] Cigar products are appealing because they come in flavors, often cost less than cigarettes, and can be used for marijuana consumption. [3] The term 'cigar' represents a specific class of products in which the tobacco is wrapped in tobacco leaf or another tobacco-containing substance. [4] There are three broad classes of cigars: cigarillos, traditional cigars, and filtered cigars. Cigar classes vary based on product characteristics, user characteristics, price, and purchase quantity. Cigarillo users are more likely to be young adults, Black, and have a household income less than 100% of the federal poverty level, and one in five reports daily use. [5,6] Cigarillos are mid-sized, often used for marijuana consumption, and half of users purchase singles, which in 2013, cost approximately $1.15 each. [3,7] Traditional cigar users are typically older, White, and have higher incomes, and few report daily use. [5,6] Traditional cigars cost approximately $1.30 per cigar, and almost half of users purchase singles. [7] Filtered cigar users are more likely to be older and have a household income less than 100% of the federal poverty level, and about one-third report daily use. [5–7] Filtered cigars are similar to cigarettes in size, shape, and use, with over half of users purchasing 20-packs which cost approximately $3.40 per pack, or $0.17 per cigar. [5–7]

Within the US, cigarettes are required to be sold in packages of 20 or more. [8] Cigars do not have a similar requirement. Cigar packages come in at least 12 different pack quantities ranging from singles to 60-packs. [9] In 2008, 5-packs accounted for over 42% of the market share, but by 2015, 2–3 packs were more common, accounting for 40% of all cigar sales. [10] In 2016, Congress passed the "Deeming Rule" which gave the Food and Drug Administration (FDA) regulatory authority of the manufacturing, distribution, and marketing of cigars. [11] As part of this regulatory authority, the FDA can set minimum and maximum pack quantity regulations for cigars. Additionally, over 120 local jurisdictions have enacted policies mandating minimum cigar pack quantities of either 4 or 5. [12–14] However, it remains unclear how pack quantity influences cigar user behavior.

It is widely acknowledged in the cigarette literature that smaller pack quantities reduce barriers to use as they are easier to conceal and carry and less expensive. [15–19] Thus, mandating a minimum pack quantity may reduce use. However, cigarette pack quantity is positively associated with consumption, such that reported daily consumption reflects the available pack quantities. [20–23] In other words, smokers tend to report smoking in quantities of 20 cigarettes per day (a pack), or 10 cigarettes per day (half a pack). Additionally, an online experiment found consumers opted for premium-priced packs of 10 cigarettes (instead of 20-packs with large discounts) to constrain their product use. [24] Therefore, it is possible mandating a minimum pack quantity could result in unintended consequences (e.g., an increase in cigar use). Importantly, cigar-specific data are needed prior to the FDA enact a minimum pack size policy.

One previous study examining cigar pack quantity using Waves 1 and 2 of the Population Assessment of Tobacco and Health (PATH) study estimated the average price paid for each cigar type. [7] They also examined cross-sectionally whether package quantity is associated

with cigar use. Notably, Persoskie and colleagues [7] excluded cigar users who used marijuana, which represent nearly half of cigar users. While use patterns may differ, it is important to consider these individuals as any cigar regulations might impact their behavior. Additionally, mostly absent from previous studies is whether average unit price and purchasing cigars by the box/pack or single influence the number of cigars used per day. We extend cigar quantity literature by examining Waves 1–3 of the PATH study, the relationship between purchase price and the number of cigars used per day, purchasing by the box or pack versus singles, and including marijuana users. Specifically, we examined whether cigar use was associated with three purchasing behaviors: (1) purchasing cigars by the box/pack or single, (2) purchase quantity, and (3) price paid per cigar.

## Methods

### Data source

We used data from the Population Assessment of Tobacco and Health (PATH) Study, a nationally representative, longitudinal cohort study. [25] The PATH Study is the first large joint research effort administered by the National Institutes of Health (NIH) and the FDA's Center for Tobacco Products to assess tobacco use and how it impacts the health of civilians nationally. The target population is the civilian household population 12 and older in all 50 US states and the District of Columbia. The PATH study used a four-stage stratified area probability sample design in Wave 1, which included a stratified sample of geographical primary sampling units, smaller geographical segments, residential addresses, and households. The first wave of the PATH study was conducted between September 12, 2013, and December 14, 2014. Institutional Review Board approval and informed consent were obtained by Westat. A total of 32,320 adults, 13,651 youth, and 13,588 parents of youth were interviewed. All Wave 1 respondents were eligible and interviewed in approximately the same month in in the later, annual waves, as long as they were still living in the U.S. and were not incarcerated. As part of the complex survey design, survey weights were designed to compensate for variable probabilities of selection, differential nonresponse rates, and possible deficiencies in the sampling frame. Survey weights also account for sampling design factors such as the stratification and sampling of primary sampling units and area segments, and the use of oversampling and nonresponse adjustment factors. Wave 3 weights were used in the present analyses, using the public-use files. [26]

### Study sample

Adults 18 and older who reported current use of any type of cigars (cigarillos, traditional cigars, and filtered cigars, including with marijuana) were included in the study sample. Current use was defined as reporting cigar use some days or every day. Users of each type of cigar were not mutually exclusive as one of the current users of one type (e.g., of cigarillos) may also be current users of the other cigar products (e.g., of traditional cigars or filtered cigars). Therefore, 3,051 cigarillo users, 2,586 traditional cigar users, and 1,295 filtered cigar users were included in the analyses.

### Measures

**Dependent variable.** The primary dependent variable was the average number of cigars used per day, analyzed as a continuous measure. The average number of cigars used was derived from the survey questions "On average, about how many [cigar product] do you now smoke each day?" and "On average, on days you smoked, how many [cigar product] did you

usually smoke each day?" Similar questions were used to assess use of cigarillos, traditional cigars, and filtered cigars.

**Independent variables.** The primary independent variables were three measurements of purchasing behavior: (1) purchasing cigars by the box/pack or single cigars, (2) purchase quantity, and (3) price paid per cigar. For each cigar type, participants were asked to report whether they usually purchase by the box or pack, or a single cigar, and the price usually paid for the unit. Participants who usually purchase cigars by the box or pack reported the number of cigars that came in the box or pack they usually buy. The average price for a cigar was then calculated by dividing the total price by package quantity.

Additional sociodemographic characteristics including age (18 to 24, 25 to 34, 35 to 44, 45 to 54, and 55 years old or older), sex (male or female), race/ethnicity (non-Hispanic White, non-Hispanic Black, Hispanic, or non-Hispanic other), income level (< 100% of poverty guideline, 100–199% of poverty guideline, and ≥ 200% of poverty guideline; calculated based on $11,770 per individual), education level (less than high school, high school graduate or equivalent, some college (no degree) or associate degree, bachelor's degree, or advanced degree), and residential region (Northeast, Midwest, South, West) were collected in the Wave 1 survey. Specific wordings and calculations for each of the items are available in the Wave 1 Adult Codebook, available at https://www.icpsr.umich.edu/icpsrweb/NAHDAP/studies/36498/datadocumentation#.

## Analyses

Sociodemographic characteristics were reported for each type of current cigar users (cigarillos, traditional cigars, and filtered cigars). The unweighted count and frequency were reported for all categorical variables, whereas weighted mean and standard deviation were reported for continuous variables. Spearman correlations were estimated to assess the association between the average cigar price and the number of cigars in a box. For each cigar type, a generalized estimating equation (GEE) model was used to examine the population-averaged (marginal) effects of purchasing behavior on cigars used per day. We used the GEE framework because the population-averaged response for the specific purchasing behavior "is directly estimable from observations without assumptions about the heterogeneity across individuals in the parameters." [27] Separate analyses were performed using the three measures of purchasing behavior. All models adjusted for sociodemographic characteristics, including age, sex, race, income level, education level, and residential region. All tests were two-sided, a p-value < 0.05 was considered significant, and all analyses were conducted in 2019 using SAS 9.4 and R 3.5.

In sensitivity analyses, we excluded participants who reported only using cigars as blunts (i.e., filled with marijuana) at Wave 1, and those who used only blunts at Wave 2 or Wave 3 were considered non-current cigar users. Additionally, analysis was restricted to those who completed all three waves with the longitudinal wave sampling weights of Wave 3 used. The significant findings and directions of the base model (analysis with all participants) did not differ from either of the sensitivity analyses; therefore, only the former data are presented.

## Results

### Sample characteristics

Table 1 presents the demographic characteristics of Wave 1 current cigarillo, traditional cigar, and filtered cigar users. Of 3,051 Wave 1 current *cigarillo* users, approximately 46.3% were 18 to 24, 69% were male, 84.0% were at least high school graduates or equivalent, and nearly half (48.8%) had a family income below 100% of the federal poverty guideline. Most current cigarillo users (89%) were everyday users, 51.9% usually purchased singles, and on average, users

**Table 1. Demographic characteristics of Wave 1 current cigarillo, traditional cigar, and filtered cigar users.**

|  | Cigarillos N = 3,051 | Traditional Cigars N = 2,586 | Filtered Cigars N = 1,295 |
|---|---|---|---|
| **Age** |  |  |  |
| 18–24 | 1,413 (46.3) | 681 (26.3) | 440 (34.0) |
| 25–34 | 677 (22.2) | 599 (23.2) | 241 (18.6) |
| 35–44 | 433 (14.2) | 460 (17.8) | 203 (15.7) |
| 45–54 | 306 (10.0) | 400 (15.5) | 190 (14.7) |
| 55–64 | 169 (5.5) | 304 (11.8) | 159 (12.3) |
| 65+ | 53 (1.7) | 142 (5.5) | 62 (4.8) |
| **Sex** |  |  |  |
| Male | 2,116 (69.4) | 2,256 (87.3) | 896 (69.2) |
| Female | 935 (30.6) | 329 (12.7) | 399 (30.8) |
| **Race/Ethnicity** |  |  |  |
| Non-Hispanic White | 1,485 (48.8) | 1,742 (67.6) | 782 (60.5) |
| Non-Hispanic Black | 779 (25.6) | 282 (10.9) | 194 (15.0) |
| Hispanic | 514 (16.9) | 337 (13.1) | 199 (15.4) |
| Non-Hispanic other | 268 (8.8) | 217 (8.4) | 118 (9.1) |
| **Region** |  |  |  |
| Northeast | 407 (13.3) | 456 (17.6) | 182 (14.0) |
| Midwest | 762 (25.0) | 630 (24.4) | 342 (26.4) |
| South | 1,308 (42.9) | 951 (36.8) | 484 (37.4) |
| West | 574 (18.8) | 549 (21.2) | 287 (22.2) |
| **Education level** |  |  |  |
| Less than high school | 485 (16.0) | 256 (10.0) | 222 (17.3) |
| High school graduate or equivalent | 1,125 (37.1) | 692 (26.9) | 460 (35.8) |
| Some college or associates degree | 1,127 (37.1) | 946 (36.8) | 469 (36.5) |
| Bachelor's degree and above | 299 (9.8) | 675 (26.3) | 134 (10.4) |
| **Income level** |  |  |  |
| < 100% of poverty guideline | 1,377 (48.8) | 699 (29.1) | 586 (49.2) |
| 100–199% of poverty guideline | 652 (23.1) | 464 (19.3) | 307 (25.8) |
| ≥ 200% of poverty guideline | 791 (28.0) | 1,240 (51.6) | 297 (25.0) |
| **Cigar use status** |  |  |  |
| Every day | 2,723 (89.2) | 167 (6.5) | 1,073 (82.9) |
| Some days | 328 (10.8) | 2,419 (93.5) | 222 (17.1) |
| **Purchase preference** |  |  |  |
| Box or pack | 581 (48.1) | 315 (40.7) | 437 (83.1) |
| Single | 627 (51.9) | 459 (59.3) | 89 (16.9) |
| **Purchase quantity** | 3.4 (0.1) | 4.4 (0.2) | 14.5 (0.3) |
| **Total purchase price** | $2.20 (0.09) | $4.98 (0.30) | $3.48 (0.20) |
| **Price per cigar** | $1.17 (0.07) | $2.73 (0.22) | $0.72 (0.19) |
| **Average number of cigars used** | 2.2 (0.05) | 2.2 (0.1) | 4.5 (0.3) |

Count and weighted frequency are reported for all categorical variables; mean and standard deviation are reported for continuous variables.

smoked 2.2 cigarillos on days they smoked during the past 30 days before the survey. Among 2,586 *traditional cigar* users, 87% were male and approximately half (51.6%) had a family income of at least 200% of federal poverty guidelines. Around 93% of current traditional cigar users smoked traditional cigars some days, 41% purchased traditional cigars by the box or pack, and on average, users smoked 2.3 traditional cigars on days they smoked during the past

**Table 2.** The association between cigar purchase patterns (by the box vs single and quantity) and the average number of cigars smoked per day.

| | Purchasing by the box/pack vs single | | | Purchasing quantity | | |
|---|---|---|---|---|---|---|
| | **Cigarillos** | **Traditional cigars** | **Filtered cigars** | **Cigarillos** | **Traditional cigars** | **Filtered cigars** |
| **B** | 1.02 | 1.40 | 2.55 | 0.16 | 0.14 | **0.24** |
| **SE(β)** | 0.12 | 0.23 | 0.87 | 0.02 | 0.01 | **0.04** |
| **95% CI** | (0.77, 1.26) | (0.95, 1.85) | (0.84, 4.25) | (0.11, 0.21) | (0.03, 0.06) | (0.16, 0.32) |
| **p-value** | <.0001 | <.0001 | <.01 | <.0001 | <.0001 | <.0001 |

p<0.05 is presented in bold. Generalized estimating equation (GEE) models were used to examine the association between (1) purchasing by the box/pack versus singles (referent group) and the average number of cigars smoked per day and (2) purchase quantity and average number of cigars smoked per day. The models adjusted for sociodemographic characteristics, including age, sex, race, income level, education level, and residential region.

30 days before the survey. Among 1,295 *filtered cigar* users, 34% were 18–24, 69% were male, and nearly half (49.2%) had a family income below 100% of the federal poverty guidelines. Around 83% of current filtered cigar users smoked filtered cigars every day, 83% purchased filtered cigars by the box or pack, and on average, users smoked 4.5 filtered cigars on days they smoked during the past 30 days before the survey.

## Pack quantity and use

The association between purchasing cigars by the box or single, and the number of cigars smoked per day is presented in Table 2. On average, cigarillo users who purchased cigarillos by the box or pack smoked 1.0 more cigarillos per day than users who purchased single cigarillos, (β = 1.02, p<0.0001). Similar results were found among traditional cigar users. Cigar users who purchased traditional cigars by the box or pack smoked 1.4 more per day than those purchasing single traditional cigars (β = 1.40, p<0.0001). Similarly, filtered cigar users who purchased filtered cigars by the box or pack smoked 2.6 more filtered cigars per day than those purchasing single filtered cigars (β = 2.55, p<0.01).

Table 2 also presents the association between the quantity of cigars in a box and the number of cigars smoked per day. A significant positive association was found between the quantity of cigars in a box and the number of cigars used among users of all three cigar types (cigarillos β = 0.16, p<0.0001; traditional cigars β = 0.04, p<0.0001; filtered cigars β = 0.24, p<0.0001).

## Pack quantity and price

We found a negative correlation between the number of cigars in a box and the average price of cigars for all three cigar types (Fig 1: cigarillos r = -0.54, p<0.001; traditional cigars r = -0.40, p<0.001; filtered cigars r = -0.77, p<0.001). After adjusting for covariates, we found that higher average price per cigar was associated with a lower number of cigars used (Table 3: traditional cigars β = -0.12, p<0.01; filtered cigars β = -0.86, p = 0.02), but an insignificant positive association was found among cigarillo users (Table 3: β = 0.08, p = 0.62).

## Discussion

This study provides further evidence on the associations between cigar pack quantity, price, and use, using data from the nationally-representative PATH study. Our longitudinal examination of Wave 1 cigar users and their cigar use across Waves 2 and 3, found that cigar pack quantity and price paid were associated with use for each cigar type, including marijuana users.

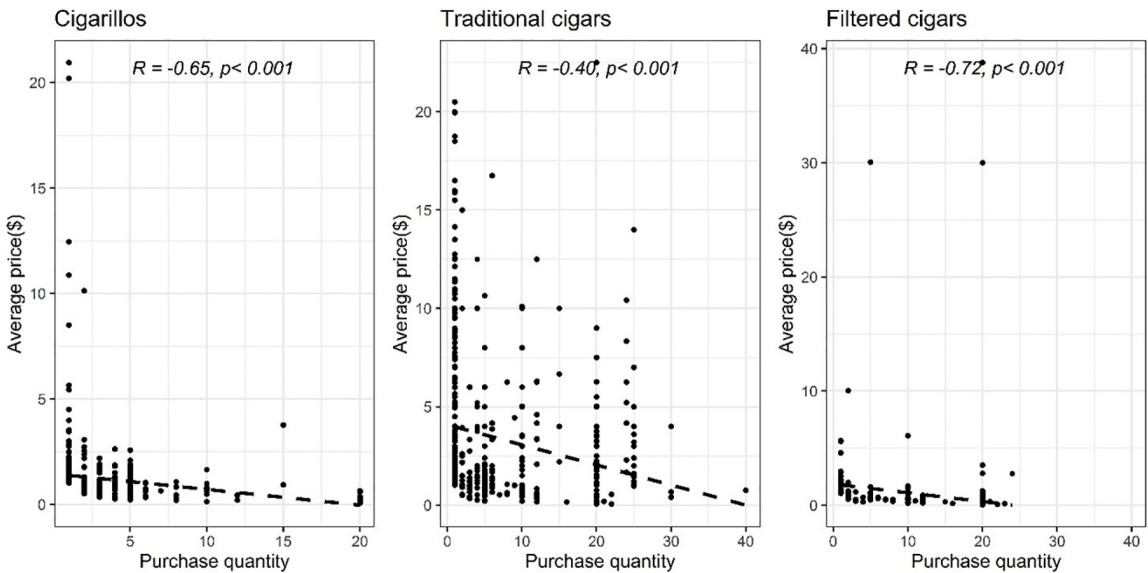

**Fig 1. Spearman's correlation between average price per cigar and purchase quantity.**

We extend Persoskie and colleagues' [7] analyses of Waves 1–2 of PATH data that identified a relationship between cigar price and pack quantity, such that as pack quantity increased, price per cigar decreased. We identified this relationship for all three cigar types, while including marijuana users and an additional wave of data. This is consistent with tobacco industry documents and general marketing strategies that indicate quantity discounts optimize purchasing. [19,28,29] This strategy has been shown to appeal to large-quantity cigarette smokers. It also likely reflects a segment of users who will pay higher for lower quantities as a strategy to self-ration their consumption.

Cigar pack purchase quantity was consistently associated with use, such that purchasing smaller quantities was associated with smoking fewer cigars per day. For each cigar type, cigar users who purchased singles smoked fewer cigars per day than those who purchased cigars by the box or pack. Additionally, in comparing purchase quantity, those who purchased smaller quantities smoked nearly half as many cigars per day. These findings suggest caution to those who advocate for minimum pack quantities, even small pack sizes of 4–5, as most communities have done. [12–14] Approximately half of cigarillo and traditional cigar smokers purchase singles, which is associated with smoking less per day. While the goal of a minimum pack size would be to reduce use, requiring users to purchase greater quantities could lead to increased use.

**Table 3. The association between price per cigar and the average number of cigars smoked per day.**

| Price per cigar | Cigarillos | Traditional Cigars | Filtered Cigars |
|---|---|---|---|
| **B** | 0.08 | -0.12 | -0.86 |
| **SE(β)** | 0.16 | 0.04 | 0.36 |
| **95% CI** | (-0.23, 0.39) | (-0.19, -0.05) | (-1.56, 0.17) |
| **p-value** | 0.62 | <0.01 | 0.02 |

p<0.05 is presented in bold. A generalized estimating equation (GEE) model was used to examine the association between price and average number of cigars smoked per day. The model adjusted for sociodemographic characteristics, including age, sex, race, income level, education level, and residential region.

As price per cigar increased, cigar use decreased for traditional cigars and filtered cigars, but not cigarillos. This provides support for the primary position of many advocating for minimum pack laws and extends cigarette literature showing increases in price are associated with decreases in use. [30–32] We are unsure why we did not identify this trend among cigarillo users. Compared to traditional and filtered cigars, cigarillo total package price, price per cigar, and purchase quantity were lower. This may be associated with price promotions common among cigarillo products, or the overall low price point seen for this product. [33] Thus, it may be important to also consider minimum price laws alongside minimum pack size laws, as some jurisdictions have done. [12]

There are distinct sociodemographic and use behavior differences across cigar type groups. For example, most filtered cigar users purchased by the box, and the average purchase quantity was three times as high as for traditional cigars or cigarillos. Traditional cigar users were more likely male, higher income, and smoked less often compared to filtered cigar or cigarillo users. However, our results for pack quantity and use were generally consistent across cigar types, even with these and other differences in patterns of use and sociodemographic characteristics of users. Despite this, we caution against assuming a policy broadly addressing cigar minimum pack quantity would impact users of each cigar type similarly. For example, a minimum pack size of ten would still be less than the current quantity purchased by most filtered cigar users, but it would more than double the typical purchase quantities for cigarillo and traditional cigar users. Additional research is needed to understand potential differential policy impact by cigar type.

Findings should be considered with regard to several limitations. The data were self-reported which may result in social desirability biased responses for use, misclassification of cigar type, or recall biases regarding purchase price or quantity. However, the PATH survey included pictures for each of the cigars and when possible obtained the actual product from participants, each of which reduces the likelihood for self-reported error. Additionally, we did not examine other product characteristics that may contribute to price, pack quantity, and use, including cigar length, weight, or flavors. Finally, because only current users were asked about pack quantity, we were unable to assess these relationships among less frequent users. Future research should examine the impact of purchase behavior among those who smoke cigars less frequently.

## Conclusion

The current study found that cigarillo, traditional cigar, and filtered cigar pack purchase quantity and price were associated with daily cigar use. Smaller pack quantities and higher price per cigar were associated with smoking fewer cigars per day. Given the authority of the FDA and local jurisdictions over cigar pack quantity, this study provides data pertinent to potential minimum and maximum package quantity policy and regulations.

## Acknowledgments

The authors have no conflicts of interest or financial disclosures to report for this work. JLK and SA conceptualized and designed the study. LS conducted data analyses. All authors contributed to drafting the manuscript and approved the final manuscript as submitted.

## Author Contributions

**Conceptualization:** Jessica L. King, Sunday Azagba.

**Formal analysis:** Jessica L. King, Lingpeng Shan, Sunday Azagba.

Writing – **original draft:** Jessica L. King, Lingpeng Shan.

Writing – **review & editing:** Jessica L. King, Lingpeng Shan, Sunday Azagba.

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
