## [Decision Letter · Decision Letter 0]

3 Apr 2020

PONE-D-20-04899

Association between purchasing behaviors and cigar use: A longitudinal analysis of Waves 1-3 of the Population Assessment of Tobacco and Health (PATH) Study

PLOS ONE

Dear Dr King,

Thank you for submitting your manuscript to PLOS ONE. After careful consideration, we feel that it has merit but does not fully meet PLOS ONE’s publication criteria as it currently stands. Therefore, we invite you to submit a revised version of the manuscript that addresses the points raised during the review process.

The reviewers and I see merit in your work.  However, there are minor details that must be address before moving forward with your paper.  The reviewers have identified a number of passages and terms that require further elaboration. 

I also request that you reshape Tables 2 & 3 so that it becomes clear(er) that the independent variables in regressions are for 1) Purchased by the box (single purchase is the referent) and 2) the number of cigar units per package (continuous) across the three types of cigars (to be placed in columns) for Table 2.  This traditional display will help readers to quickly glean the independent and dependent variables in each analysis.  Similarly, labeling the independent variable "Price per package/unit" would also make it clear(er) for the reader. 

Along these lines, please reconsider the titles of your tables for greater clarity.  Table 2 could be re-titled, "The Association between Cigar Purchase Type and Cigars per Package on the Average Number of Cigars Smoked/Consumed per Day," or some other variant of such a title.  Please also change the title of Table 3 accordingly. 

We would appreciate receiving your revised manuscript by May 18 2020 11:59PM. To enhance the reproducibility of your results, we recommend that if applicable you deposit your laboratory protocols in protocols.io, where a protocol can be assigned its own identifier (DOI) such that it can be cited independently in the future. For instructions see: http://journals.plos.org/plosone/s/submission-guidelines#loc-laboratory-protocols

We look forward to receiving your revised manuscript.

Kind regards,

Bryan L. Sykes, Ph.D.

Academic Editor

PLOS ONE

Journal Requirements:

Please ensure that your manuscript meets PLOS ONE's style requirements, including those for file naming. The PLOS ONE style templates can be found at http://www.plosone.org/attachments/PLOSOne_formatting_sample_main_body.pdf and http://www.plosone.org/attachments/PLOSOne_formatting_sample_title_authors_affiliations.pdf

Reviewers' comments:

Reviewer's Responses to Questions

**Comments to the Author**

1. Is the manuscript technically sound, and do the data support the conclusions?

Reviewer #1: Yes

Reviewer #2: Yes

2. Has the statistical analysis been performed appropriately and rigorously? 

Reviewer #1: Yes

Reviewer #2: Yes

3. Have the authors made all data underlying the findings in their manuscript fully available?

Reviewer #1: Yes

Reviewer #2: Yes

4. Is the manuscript presented in an intelligible fashion and written in standard English?

Reviewer #1: Yes

Reviewer #2: Yes

5. Review Comments to the Author

Reviewer #1: This is a well-written and rigorous paper that makes an important contribution to tobacco regulatory science. I have a few modest suggestions and comments for the authors.

-In line 29 of the abstract, I believe that the p-values should be reported as p<.001, not p<.0001.

-Line 41, I'd specify that they "cost less THAN CIGARETTES," since it's not clear what the comparison is.

-The data source section in the methods should include citations (i.e. Hyland et al).

-Line 108- can you specify how "current use" is defined in PATH?

-In line 222 of the discussion, you state that as price per cigar decreased for cigarillos and traditional cigars, use decreased, but not for filtered cigars. I may be misunderstanding, but should this say FILTERED CIGARS and traditional cigars? Based on Table 3, it looks like the relationship between price per cigarillo and average number of cigarillos per day is not significant, but is for filtered cigars.

-Line 242- this ties back to my prior comment about the definition of current use. I think it is worth noting in the limitations that because only current users are asked about pack size, you are unable to assess the relationship between pack size/quantity among non-regular/less frequent/experimenting users of cigar products. I think that this is important given that cigars are used differently than cigarettes by consumers.

Reviewer #2: This manuscript uses data from Waves 1-3 of the PATH Study to examine associations between cigar use behavior, purchase quantity, and purchase price. This is an interesting manuscript and it addresses an important topic – how to regulate price to maximally protect public health. This is an important issue from FDA-regulatory perspective. A few comments for consideration:

Line 41: Please define what is meant by “current.” Does this mean past 30 days? Ever use during this period? Daily? Regular?

Line 49: The cost per cigarillo is estimated at $1.15. At what point in time?

Line 53-54: It be helpful to include the average price per filtered cigar, in addition to cost per pack.

Line 66-73: Relying on the cigarette literature is understandable. At the same time, can the FDA rely on the cigarette literature to inform cigar policy? Are there cigar-specific literature that can be relied upon in addition? This cigar-specific connection seems critical for FDA’s purpose.

Line 73: It seems that the implied unintended consequence is an increase in use. You might make this explicit here.

Line 108: Please define what is meant by “current.”

Linde 132: If possible, please define what is meant by “poverty guideline.”

Line 201: Is this study truly longitudinal? The PATH Study clearly has a longitudinal design, but did the current analysis measure within-person change? Or, did it compare across time (like a repeated cross-sectional study?

Line 218-219: The paper claims that most communities have advocated for small pack sizes f 4-5. Can you provide a citation to support this claim?

6. PLOS authors have the option to publish the peer review history of their article (what does this mean?). If published, this will include your full peer review and any attached files.

Reviewer #1: No

Reviewer #2: No

---

## [Author Response · Author response to Decision Letter 0]

4 May 2020

Thank you for the helpful feedback and the opportunity to improve our work. We have addressed each of the comments and detailed the actions taken below.

Response to Reviewers

1. Reshape Tables 2 & 3 so that it becomes clear(er) that the independent variables in regressions are for 1) Purchased by the box (single purchase is the referent) and 2) the number of cigar units per package (continuous) across the three types of cigars (to be placed in columns) for Table 2. This traditional display will help readers to quickly glean the independent and dependent variables in each analysis. 

We edited tables 2 and 3 as suggested, with the products by columns.

2. Similarly, labeling the independent variable "Price per package/unit" would also make it clear(er) for the reader.

We added this for Table 3, but with Table 2 there are two independent variables: 1. purchasing by the box/pack vs single, and 2. purchasing quantity. With the tables restructured as suggested, we think it makes this clearer. 

3. Please reconsider the titles of your tables for greater clarity. Table 2 could be re-titled, "The Association between Cigar Purchase Type and Cigars per Package on the Average Number of Cigars Smoked/Consumed per Day," or some other variant of such a title. Please also change the title of Table 3 accordingly.

We edited the table titles as suggested. Table 2 is now The Association between Cigar Purchase Patterns (by the Box vs Single and Quantity) and the Average Number of Cigars Smoked per Day and Table 3 is now The Association between Price per Cigar and the Average Number of Cigars Smoked per Day.

Reviewer #1: This is a well-written and rigorous paper that makes an important contribution to tobacco regulatory science. I have a few modest suggestions and comments for the authors.

1. In line 29 of the abstract, I believe that the p-values should be reported as p<.001, not p<.0001.

As written, the p-values match Tables 2 and 3. We are happy to edit if this is a stylist concern for the journal, but it’s unclear whether this comment refers to that or consistency with the tables.

For reference, the guidelines state: P-values less than 0.001 may be expressed as p < 0.001, or as exponentials in studies of genetic associations, which does not clarify whether they must be presented this way.

2. Line 41, I'd specify that they "cost less THAN CIGARETTES," since it's not clear what the comparison is.

We edited this as suggested. Line 41 now reads: Cigar products are appealing because they come in flavors, cost less than cigarettes, and can be used for marijuana consumption.

3. The data source section in the methods should include citations (i.e. Hyland et al).

We added references to Hyland and the PATH User Guide, at lines 97 and 106.

4. Line 108- can you specify how "current use" is defined in PATH?

We used survey item R01_AG1003 – “Do you now smoke []” every day, some days, or not at all

The text at Line 114 now states: Current use was defined as reporting cigar use some days or every day.

5. In line 222 of the discussion, you state that as price per cigar decreased for cigarillos and traditional cigars, use decreased, but not for filtered cigars. I may be misunderstanding, but should this say FILTERED CIGARS and traditional cigars? Based on Table 3, it looks like the relationship between price per cigarillo and average number of cigarillos per day is not significant, but is for filtered cigars.

Yes, thank you for highlighting this error. We have edited this to say filtered cigars, not cigarillos and edited the discussion that followed. The text now states: 

As price per cigar increased, cigar use decreased for traditional cigars and filtered cigars, but not cigarillos. This provides support for the primary position of many advocating for minimum pack laws and extends cigarette literature showing increases in price are associated with decreases in use.(30–32) We are unsure why we did not identify this trend among cigarillo users. Compared to traditional and filtered cigars, cigarillo total package price, price per cigar, and purchase quantity were lower. This may be associated with price promotions common among cigarillo products, or the overall low price point seen for this product.(33) Thus, it may be important to also consider minimum price laws alongside minimum pack size laws, as some jurisdictions have done.(12) 

6. Line 242- this ties back to my prior comment about the definition of current use. I think it is worth noting in the limitations that because only current users are asked about pack size, you are unable to assess the relationship between pack size/quantity among non-regular/less frequent/experimenting users of cigar products. I think that this is important given that cigars are used differently than cigarettes by consumers.

We added this as a limitation as suggested. The text at Lines 262-264 now states: Finally, because only current users were asked about pack quantity, we were unable to assess these relationships among less frequent users. Future research should examine the impact of purchase behavior among those who smoke cigars less frequently. 

Reviewer #2: This manuscript uses data from Waves 1-3 of the PATH Study to examine associations between cigar use behavior, purchase quantity, and purchase price. This is an interesting manuscript and it addresses an important topic – how to regulate price to maximally protect public health. This is an important issue from FDA-regulatory perspective. A few comments for consideration:

1. Line 41: Please define what is meant by “current.” Does this mean past 30 days? Ever use during this period? Daily? Regular?

This citation is from an MMWR report that used the 2017 NHIS data. The questionnaire is here: file:///C:/Users/u6026807/Downloads/qadult.pdf. Based on the description in the methods, it appears the authors used this item: Do you now smoke regular cigars, cigarillos, or little filtered cigars every day, some days, or not at all? Every day, some days, not at all. We agree this is not very specific and is open to interpretation. This is, however, an item typically used, and, unfortunately, we are unable to provide additional details, as this is the language used within the report. 

2. Line 49: The cost per cigarillo is estimated at $1.15. At what point in time?

We added this detail, as suggested. The text now states: Cigarillos are mid-sized, often used for marijuana consumption, and half of users purchase singles, which in 2013, generally cost approximately $1.15 each.

3. Line 53-54: It be helpful to include the average price per filtered cigar, in addition to cost per pack.

We added this as suggested. The text now states: Filtered cigars are similar to cigarettes in size, shape, and use, with over half of users purchasing 20-packs which cost approximately $3.40 per pack, or $0.17 per cigar.

4. Line 66-73: Relying on the cigarette literature is understandable. At the same time, can the FDA rely on the cigarette literature to inform cigar policy? Are there cigar-specific literature that can be relied upon in addition? This cigar-specific connection seems critical for FDA’s purpose.

There is very little cigar-specific data. The present study and the noted Persoskie study are the two specific ones of which we are aware. We added a sentence noting the importance of cigar-specific data to enact a policy. The text at Line 74 states: Importantly, cigar-specific data are needed for the FDA to enact a cigar policy.

5. Line 73: It seems that the implied unintended consequence is an increase in use. You might make this explicit here.

We made this explicit, as suggested. The text now states: Therefore, it is possible mandating a minimum pack quantity could result in unintended consequences (e.g., an increase in cigar use).

6. Line 108: Please define what is meant by “current.”

We added details on the specific items used. The text at Line 114 now states: Current use was defined as reporting cigar use some days or every day.

7. Linde 132: If possible, please define what is meant by “poverty guideline.”

The PATH Study derives this measure using a 4 step process:

STEP 1: Estimate household-level income from the mid-point of each household income range reported in R01_AM0030. For income ranges higher than $100,000, categories are collapsed and an effective household income of $120,000 is assigned.

STEP 2: Calculate the poverty income guideline based on the 2015 POVERTY GUIDELINES FOR THE 48 CONTIGUOUS STATES AND THE DISTRICT OF COLUMBIA for the household as (11770 + ((Aggregate count of people in household from screener - 1)* 4160)).

STEP 3: Calculate family income as a percentage of the HHS poverty guideline. Poverty variable calculation formula: [Effective family income] / [poverty guideline] * 100 = [family income as a percentage of the HHS poverty guideline]. To be at 100% of the poverty guideline is equivalent to having a family income that is the same as the HHS-specified poverty guideline. A level less than 100% indicates having a family income less than the poverty guideline and a level greater than 100% indicate having a family income greater than the poverty guideline.

STEP 4: IF family income as a percentage of the HHS poverty guideline < 100, THEN R01R_POVCAT3 = 1; IF family income as a percentage of the HHS poverty guideline >= 100 and < 200, THEN R01R_POVCAT3 = 2; IF family income as a percentage of the HHS poverty guideline >= 200, THEN R01R_POVCAT3 = 3; ELSE IF R01_AM0030 = -9 THEN R01R_POVCAT3 = -99999; ELSE IF R01_AM0030 = -8 THEN R01R_POVCAT3 = -99988; ELSE IF R01_AM0030 = -7 THEN R01R_POVCAT3 = -99977; ELSE IF R01_AM0030 = -1 THEN R01R_POVCAT3 = -99911.

We added details at Line 136: income level (< 100% of poverty guideline, 100-199% of poverty guideline, and ≥ 200% of poverty guideline; calculated based on $11,770 per individual) 

We also added a note that the full description for each of the variables is available in the codebooks and included a link. The text at Line 137 now states: Specific wordings and calculations for each of the items are available in the Wave 1 Adult Codebook, available at https://www.icpsr.umich.edu/icpsrweb/NAHDAP/studies/36498/datadocumentation#. 

8. Line 201: Is this study truly longitudinal? The PATH Study clearly has a longitudinal design, but did the current analysis measure within-person change? Or, did it compare across time (like a repeated cross-sectional study?

The study design accounted for the longitudinal design. The use of GEE relaxed the independent assumption, and to account for the within-group correlation structure, we used an exchangeable working correlation structure.

9. Line 218-219: The paper claims that most communities have advocated for small pack sizes f 4-5. Can you provide a citation to support this claim?

We added the following citations, as suggested:

Counter Tobacco. Restricting Product Packaging – Counter Tobacco [Internet]. 2019 [cited 2019 Sep 30]. Available from: https://countertobacco.org/policy/restricting-product-packaging/

Li W, Gouveia T, Sbarra C, Harding N, Kane K, Hayes R, et al. Has Boston’s 2011 cigar packaging and pricing regulation reduced availability of single-flavoured cigars popular with youth? Tob Control. 2017 Mar 1;26(2):135–40. 

Sbarra C, Reid M, Harding N, Li W. Promising Strategies to Remove Inexpensive Sweet Tobacco Products From Retail Stores. Public Health Rep. 2016 Dec 12;132(1):106–9.

---

## [Decision Letter · Decision Letter 1]

17 Jun 2020

Association between purchasing behaviors and cigar use: A longitudinal analysis of Waves 1-3 of the Population Assessment of Tobacco and Health (PATH) Study

PONE-D-20-04899R1

Dear Dr. King,

We’re pleased to inform you that your manuscript has been judged scientifically suitable for publication and will be formally accepted for publication once it meets all outstanding technical requirements.

Kind regards,

Bryan L. Sykes, Ph.D.

Academic Editor

PLOS ONE

Additional Editor Comments (optional):

Reviewers' comments:

Reviewer's Responses to Questions

**Comments to the Author**

1. If the authors have adequately addressed your comments raised in a previous round of review and you feel that this manuscript is now acceptable for publication, you may indicate that here to bypass the “Comments to the Author” section, enter your conflict of interest statement in the “Confidential to Editor” section, and submit your "Accept" recommendation.

Reviewer #1: All comments have been addressed

Reviewer #2: All comments have been addressed

2. Is the manuscript technically sound, and do the data support the conclusions?

Reviewer #1: Yes

Reviewer #2: Yes

3. Has the statistical analysis been performed appropriately and rigorously? 

Reviewer #1: Yes

Reviewer #2: Yes

4. Have the authors made all data underlying the findings in their manuscript fully available?

Reviewer #1: Yes

Reviewer #2: Yes

5. Is the manuscript presented in an intelligible fashion and written in standard English?

Reviewer #1: Yes

Reviewer #2: Yes

6. Review Comments to the Author

Reviewer #1: The authors have sufficiently responded to my comments and questions. This is an well-written paper and an important contribution to tobacco regulatory science.

Reviewer #2: (No Response)

7. PLOS authors have the option to publish the peer review history of their article (what does this mean?). If published, this will include your full peer review and any attached files.

Reviewer #1: No

Reviewer #2: No

---

## [Editor Report · Acceptance letter]

19 Jun 2020

PONE-D-20-04899R1 

Association between purchasing behaviors and cigar use: A longitudinal analysis of Waves 1-3 of the Population Assessment of Tobacco and Health (PATH) Study 

Dear Dr. King:

I'm pleased to inform you that your manuscript has been deemed suitable for publication in PLOS ONE. Congratulations! Your manuscript is now with our production department. 

Kind regards, 

on behalf of

Dr. Bryan L. Sykes 

Academic Editor

PLOS ONE